# The Photoperiod-Driven Cyclical Secretion of Pineal Melatonin Regulates Seasonal Reproduction in Geese (*Anser cygnoides*)

**DOI:** 10.3390/ijms241511998

**Published:** 2023-07-26

**Authors:** Qiang Bao, Wang Gu, Lina Song, Kaiqi Weng, Zhengfeng Cao, Yu Zhang, Yang Zhang, Ting Ji, Qi Xu, Guohong Chen

**Affiliations:** 1Key Laboratory for Evaluation and Utilization of Poultry Genetic Resources of Ministry of Agriculture and Rural Affairs, Yangzhou University, Yangzhou 225009, China; dx120200144@stu.yzu.edu.cn (Q.B.); dx120210144@stu.yzu.edu.cn (W.G.); sln990907@163.com (L.S.); 18705279093@163.com (K.W.); caozhengfeng1991@163.com (Z.C.); yuzhang@yzu.edu.cn (Y.Z.); zyang@yzu.edu.cn (Y.Z.); tji@yzu.edu.cn (T.J.); 2Joint International Research Laboratory of Agriculture and Agri-Product Safety, The Ministry of Education of China, Yangzhou University, Yangzhou 225009, China

**Keywords:** goose, photoperiod, hypothalamus–pituitary–gonadal axis, pineal gland, seasonal reproduction

## Abstract

The photoperiod is the predominant environmental factor that governs seasonal reproduction in animals; however, the underlying molecular regulatory mechanism has yet to be fully elucidated. Herein, Yangzhou geese (*Anser cygnoides*) were selected at the spring equinox (SE), summer solstice (SS), autumn equinox (AE), and winter solstice (WS), and the regulation of seasonal reproduction via the light-driven cyclical secretion of pineal melatonin was investigated. We show that there were seasonal variations in the laying rate and GSI, while the ovarian area decreased 1.5-fold from the SS to the AE. Moreover, not only did the weight and volume of the pineal gland increase with a shortened photoperiod, but the secretory activity was also enhanced. Notably, tissue distribution further revealed seasonal oscillations in melatonin receptors *(Mtnrs*) in the pineal gland and the hypothalamus–pituitary–gonadal (HPG) axis. The immunohistochemical staining indicated higher Mtnr levels due to the shortened photoperiod. Furthermore, the upregulation of aralkylamine *N*-acetyltransferase (*Aanat*) was observed from the SS to the AE, concurrently resulting in a downregulation of the gonadotrophin-releasing hormone (*GnRH*) and gonadotropins (*GtHs*). This trend was also evident in the secretion of hormones. These data indicate that melatonin secretion during specific seasons is indicative of alterations in the photoperiod, thereby allowing for insight into the neuroendocrine regulation of reproduction via an intrinsic molecular depiction of external photoperiodic variations.

## 1. Introduction

Biological rhythms are an inherent property of organisms that allow for living systems to compartmentalize and carry out cellular processes, physiological signals, and behaviors with the predictable rhythms of their environment [1,2,3]. Among these, the seasonal reproduction rhythm is an evolutionary adaptation strategy for animals living in areas with environmental changes, which is thought to be essential for breeding success and for the survival of the offspring [4,5,6]. Seasonally breeding animals, especially females, are characterized by physiological changes, growth, and development [7]. Typically, animals display timely physiological changes between being reproductively active and taking a sexual rest, ensuring that births are synchronized at the most favorable season [8]. According to previous reports, the annual cycle of these switches depends upon major central regulations of the neuroendocrine system, but they are primarily orchestrated at the hypothalamus–pituitary–gonadal (HPG) axis level through the environmental control of hormone release [9,10,11]. Therefore, it is crucial to find an optimal time for the HPG axis to be activated in seasonally breeding animals. Such complex underlying mechanisms facilitate and contribute to the precise regulation of seasonal reproduction, which allows for animals to begin sexual activity at the perfect season.

Although weather parameters in different seasons are widely variable in terms of environmental factors such as temperature, humidity, precipitation, and wind speed, the photoperiod (the length of the light period) is considered to be the underlying factor used to detect seasonal variations for many organisms, including higher plants [12], insects [13], and vertebrates [8,11]. In recent years, a limited number of studies on the control of seasonal reproduction behaviors (such as courtship and mating) in mammals have explored the dynamic relationships between the photoperiod and the reproductive endocrine system [8]. According to previous studies, external photoperiodic signals are believed to stimulate the production of critical regulators in mammalian reproduction [14]. Among these, the gonadotropin-releasing hormone (GnRH), which is synthesized and secreted in the hypothalamus, is a critical neuroendocrine regulator of reproductive function in vertebrates [15]. Numerous studies have found that reproductive function is coordinated by kisspeptin (Kiss) neurons and the GnRH [16,17]. The hypothalamus integrates signals from external stimuli to generate Kiss, a ligand of the G-protein-coupled receptor GPR54, which, in turn, activates GnRH neurons to initiate the production of the GnRH [18]. In the literature about avian species, however, the photoperiodic control of kisspeptin is relatively scarce and controversial. Moreover, it was demonstrated that the GnRH was able to control the secretion of hypophyseal gonadotropins (GtHs) such as the follicle-stimulating hormone (FSH) and luteinizing hormone (LH) from the pituitary gland, and these, in turn, induce the secretion of estradiol (E2) and progesterone (P4) [11,19]. These hormones were described to be involved in follicular growth and ovulation. Therefore, such changes in the neuroendocrine homeostasis are likely to play a key role in the seasonal control of reproductive function. Compared to other vertebrates, birds may have highly sophisticated regulation mechanisms of the photoperiod, thus changing the HPG axis activity in different seasons [20]. Several studies have found that the hypothalamic suprachiasmatic nucleus (SCN) serves as the primary photoreceptive system in most vertebrates, whereas birds possess a photoreceptive system that encompasses the pineal gland and SCN [21,22,23]. Notably, the pineal gland assumes a crucial role in modulating the annual rhythm through light signal transmission. However, our current knowledge about the photoperiodic regulation of seasonal reproduction in birds remains limited.

Seasonal breeding animals can be divided into two categories according to the length of the photoperiod [11]. Currently, most birds are called long-day breeders because they develop gonads during a long photoperiod (LP), while a short photoperiod (SP) inhibits their reproductive behaviors [5]. The generally accepted hypothesis states that most of these processes are ultimately controlled by altering the signal output from the endocrine cells in the pineal gland [5,14]. The pineal gland in birds, unlike that in mammals, is a functional photoreceptive organ [20]. External light signals, particularly the changes in the photoperiod at different seasons of the year, affect melatonin (*N*-acetyl-5-methoxytryptamine) synthesis and secretion in the pineal gland [24]. Some studies reported that melatonin can provide the photoperiodic information for the control and timing of life activities in the ontogenetic process such as growth and development, mating, and reproduction [14,25]. Thus, there has been much speculation about the regulation of gonadal development and reproductive function via the pineal gland through sensing and responses to environmental changes. In addition, melatonin synthesized by the pineal gland is a highly conserved neurohormone, which is mainly limited by the activity of the *N*-acetyltransferase (Aanat, EC 2.3.1.87) [26]. According to some previous studies, melatonin is also highly lipid-soluble and readily traverses the cell membranes and all morphophysiological barriers such as the blood–brain barrier, exerting its physiological effects through its ubiquitously distributed receptors [27,28,29]. Among avian species, this view is further supported by the fact that expression levels of melatonin receptors (*Mtnrs*) were detected in the hypothalamus paraventricular nucleus [30]. Up to now, three Mtnr subtypes have been characterized, melatonin receptor 1A (*Mtnr1A*, *Mel-1a*), melatonin receptor 1B (*Mtnr1B*, *Mel-1b*), and melatonin receptor 1C (*Mtnr1C*, *Mel-1c*), which belong to the superfamily of G-protein-coupled receptors [30,31]. *Mel-1a* and *Mel-1b* are present in humans and other mammals, while *Mel-1c* as an additional melatonin receptor subtype has been identified in fish, amphibians, and birds [32]. Although melatonin plays a key role in different organs for the timing of the breeding season [8], the specific molecular and cellular mechanisms are still elusive. To summarize, relative to mammals, information about the regulation of reproduction seasonality in domestic poultry, particularly in neuroendocrine levels, is poorly understood. In this context, the regulation of the annual (i.e., seasonal) rhythm of birds needs to be further investigated.

Geese are known as animals with a strong seasonal reproduction pattern. They typically lay eggs in the winter and spring, while refraining from breeding during the summer and autumn seasons. Given the role of melatonin secreted by the pineal gland in seasonal reproduction, we explored a hypothesis that the pineal gland directly senses the changes in the length of light throughout the year, which guide its activity to regulate the cyclical secretion of melatonin, thereby controlling the gonadal development and seasonal reproduction in geese. Overall, these results contribute to a better understanding of the physiological regulatory mechanism underlying seasonal reproduction of domestic geese.

## 2. Results

### 2.1. Seasonal Photoperiods Affect the Reproductive Cycle and Gonad Development

Most seasonal changes in photoperiods and laying rates of the female Yangzhou geese are shown in Figure 1. These geese are among Chinese indigenous domestic goose breeds, mainly located in the north of Jiangsu Province in China (Figure 1A), and the natural photoperiods at the sampling site exhibited marked seasonal fluctuation (10.07–14.24 h). Their seasonal cycle of laying rates showed changes due to fluctuations in the annual photoperiod (Figure 1B). The laying rates gradually increased from January (8.66%) to February (29.37%); the highest levels were observed in March (38.70%), slowly decreasing until May (23.41%); they then stopped reproducing in June and started laying eggs again during the final 2 months of the year. Moreover, we measured the egg-laying time during the 24 h day at the SE and WS (Figure 1C,D). The egg-laying time of female Yangzhou geese was relatively concentrated; more than 50% of the eggs were laid from 4:00 a.m. to 8:00 a.m.

The morphology of ovarian development during the different breeding seasons further revealed dramatic seasonal changes in the reproductive performance (Figure 2). At the SE and WS, female geese had normal ovaries and many hierarchical follicles, while the gonads exhibited significant atrophy and degeneration from the SS to AE (Figure 2A). In addition, almost all follicles presented atresia at the AE with a sudden shortening of the photoperiod. The GSI levels significantly increased at the SE compared to WS, while, at the AE, a shortened photoperiod induced the lowest levels (*F* = 103.402, *p* < 0.001, Figure 2B). The ovarian area also exhibited a similar seasonal trend to that of the GSI, increasing from the WS to SE and shrinking from the SS to AE (*F* = 72.496, *p* < 0.001, Figure 2C).

### 2.2. Seasonal Photoperiods Affect the Morphologic Characteristics of the Pineal Gland

The morphologic characteristics of the pineal gland also showed a relatively marked seasonal variation (Figure 3A). Significant increases in volume (*F* = 25.245, *p* < 0.001) and weight (*F* = 10.338, *p* < 0.001) were observed at the AE compared to the SS (Figure 3B,C). The anatomical observations found that the pineal gland was composed of the pineal organ (PO), pineal stalk (PS), and pineal choroid (PC) (Figure 3A). The connective tissue capsule was present in the form of stripes, extending in various degrees into the parenchyma, and most blood vessels were distributed across the lateral of the pineal organ parenchyma. The number of blood vessels was lower at the SS. The pineal parenchyma was distributed with a large number of follicular structures. There was a significant increase in the number of pineal follicle-like structures from the SS to the AE (*F* = 9.822, *p* < 0.001, Table 1). In addition, compared to other breeding seasons, the long diameter of pineal follicles reached its maximum at the WS (*F* = 16.871, *p* < 0.001), while the height of follicular cells decreased significantly at the SS (*F* = 10.218, *p* < 0.001).

The ultrastructure of the pineal gland also showed the typical seasonal features (Figure 3D). Pineal cells were visibly elliptic, circular, or irregular in shape and had many small surface protrusions. The nuclei of cells were comparatively large, and abundant heterochromatin was localized near the nuclear membrane. At the SS, the micrographs showed that the cell nuclei became smaller and heterochromatin density increased. In addition, pineal gland cells contained abundant organelles. The Golgi apparatus exhibited a small area and narrow width at the SS, while being composed of many flattened cisternae arranged in stacks with numerous small vesicles at the AE. During the period from the SS to the AE, the size and number of intracellular secretory vesicles increased. Lastly, the observation of ultrastructural characteristics demonstrated prominent glycogen granules in the cytoplasm.

### 2.3. The Expression of Melatonin Biofunction-Related Genes and Endogenous Melatonin Level

*Mtnrs* showed a wide tissue distribution, being found in the pineal gland, hypothalamus, pituitary gland, and ovary (Figure 4). *Mel-1a* had an obvious seasonal fluctuation in the pineal gland and hypothalamus, and the expression level increase rapidly after the SS, reaching a higher expression at the AE (*F* = 39.577, *p* < 0.001; *F* = 49.654, *p* < 0.001, respectively) (Figure 4A,B). The expression levels of three *Mtnrs* all showed at least threefold changes at the AE compared to the values at the SS in the pituitary gland (*F* = 16.332, *p* < 0.001, Figure 4C). Among them, *Mel-1b* had a higher expression compared to other *Mtnrs* at the WS (*p* < 0.001). In the ovary, stable expressions of *Mel-1a* and *Mel-1b* were consistently at a lower level during the different seasons, while *Mel-1c* showed a 2.5-fold increase from the SS to AE (*F* = 85.114, *p* < 0.001, Figure 4D). Moreover, immunohistochemistry was used to assess the expression of Mtnr in the pineal gland. As shown in Figure 4E, Mtnrs were widely expressed in the pineal organ, with higher levels observed near the connective tissue capsule. Likewise, there were also seasonal variations in Mtnr expression levels (Figure 4F). The expression level of Mtnr decreased from the WS to SE, while it increased significantly at the AE with a shortened photoperiod compared to the SS (*F* = 28.199, *p* < 0.001). Lastly, there were consistent seasonal fluctuations between the expression levels of the *Aanat* gene and the contents of melatonin (MT). The expression of *Aanat* increased significantly with a shortening of the photoperiod from the SS to AE (*F* = 186.361, *p* < 0.001, Figure 4G). Serum melatonin levels were significantly elevated under a shorter photoperiod, reaching a maximum increase approaching threefold at the AE compared to the SS (*F* = 45.996, *p* < 0.001, Figure 4H).

### 2.4. Seasonal Photoperiods Change the Relative Expression of Reproduction-Related Genes

The results for seasonal changes in reproduction-related genes are shown in Figure 5. *GnRH* showed a highest expression level at the SE, and then its expression decreased steadily from the SS to AE with a shortened photoperiod (*F* = 50.299, *p* < 0.001, Figure 5A). In the pituitary gland, the expression levels of *FSHβ* and *LH* gradually decreased with longer photoperiods, before increasing again at the WS (Figure 5B,C). Consistently, the results of *FSHR* and *LHR* also showed similar seasonal oscillations and were expressed at the highest levels at the SE (*F* = 103.388, *p* < 0.001; *F* = 99.268, *p* < 0.001, respectively) (Figure 5E,F). On the contrary, the expression of *PRL* in the pituitary gland reached a maximum peak at the SS and showed an almost fourfold increase compared to the AE and WS (*F* = 110.401, *p* < 0.001, Figure 5D). Furthermore, the expression of *PRLR* in the ovary demonstrated a similar trend, with the lowest level at the AE and the highest at the SS (*F* = 156.698, *p* < 0.001, Figure 5G).

### 2.5. Seasonal Variation in Reproductive Hormone under the Natural Photoperiod

The levels of reproductive hormone secretion through the different seasons are shown in Figure 6. Among them, the concentrations of GnRH, FSH, LH, E2, and P4 showed similar seasonal oscillation patterns in a short photoperiod and long photoperiod. At the SE, as the photoperiod lengthened, the concentration of these hormones in serum significantly increased compared to the WS, while the lowest levels were found at the AE (*F* = 75.921, *p* < 0.001; *F* = 240.707, *p* < 0.001; *F* = 49.034, *p* < 0.001; *F* = 81.526, *p* < 0.001; *F* = 59.137, *p* < 0.001, respectively). Although the result of PRL also showed obvious fluctuations over the different seasons, the highest values corresponded to the SS (*F* = 506.974, *p* < 0.001, Figure 6D).

## 3. Discussion

The photoperiod is the main environmental factor governing seasonal changes in female Yangzhou geese, and it is directly involved in the regulation of seasonal reproduction. To uncover the regulatory mechanisms that synchronize photoperiodic cues and seasonal reproduction, we reported the anatomy, ultrastructure, and secretory activity characterizing the pineal gland under different photoperiod regimes, with a particular focus on regulation by the hypothalamus–pituitary–gonadal (HPG) axis. In the present study, the pineal gland, in response to the seasonal photoperiod, drove the alternation between the laying period (LP) and ceased-laying period (CP). The seasonal secretion of melatonin reflected changes in the photoperiod, and we thereby provided insight into neuroendocrine control of reproduction via an internal molecular representation of external photoperiod changes (Figure 7).

### 3.1. Effects of Seasonal Photoperiod on Gonadal Activity

This study observed a marked rhythm of seasonal reproduction in female Yangzhou geese, which we speculate is governed by seasonal changes in photoperiod. In the present research, the seasonal changes in photoperiod were crucial not only as a reliable environmental indicator of physiological changes but also as one of the most important environmental cues of seasonal reproduction control in vertebrate species [33,34]. During the feeding management of broilers, several researchers have successfully regulated the sexual maturation and egg production traits by changing the means of photoperiod [35,36]. In general, most studies using ambulatory recordings of seasonal photoperiodic changes have been conducted between latitudes 32° N and 51° N [11,14,37]. The site of this experimental research was in that same region, and it exhibited a significant photoperiod change of more than 4 h between the summer and winter. This meant that the location was highly suitable for exploring seasonal variations in photoperiod and the neuroendocrine control of reproduction. In seasonally breeding animals, timing of reproduction in females can influence offspring development and survival [11], and the laying performance is an important economic factor in goose production. Therefore, female Yangzhou geese were used as the subjects of this study. We found that they showed a significant increase in egg rates with prolongation of the photoperiod, after reaching the peak of laying at the SE. This is in agreement with earlier data by Yang et al. [38], where the laying rate of breeding geese was highest in February and then started decreasing. On the basis of this characteristic, the female Yangzhou geese exhibited a marked seasonal breeding pattern, similar to that found in other long-day breeders. This was also consistent with previous findings that the reproduction season of Sichuan white geese lasts for approximately 7 months [39].

In addition, the growth and development of gonads were the critical seasonal indicators affecting the reproductive performance of poultry [40,41,42]. The gonads of seasonal breeding poultry development were accompanied by annual periods of quiescence and renaissance. During the breeding season, the HPG axis of birds was activated, resulting in a significant increase in the volume of the gonads, sometimes even more than 100-fold [42,43]. Interestingly, the HPG axis could automatically shut down after the end of the breeding season, and the gonads could also degenerate [11]. In this experiment, female geese had well-developed ovaries with distinct hierarchical follicles during the peak egg-laying period at the SE with an extended photoperiod, whereas the ovaries shrank at the SS and AE, before entering the ceased-laying period. Moreover, seasonal variation in GSI was also observed in female geese. Consistent with this observation, an increase in ovarian weight was associated with greater amounts of corpora lutea, thus indicating a higher reproductive performance [44]. Meanwhile, we found the GSI levels reached their peak at the SE with a long photoperiod. Notably, we found that the gonads degenerated gradually after breeding, even though the photoperiod still increased. This phenomenon is known as photorefractoriness, and the specific molecular mechanisms regulating this phenomenon are unclear at the moment [20]. Altogether, these findings might suggest a higher sophisticated photoperiodic mechanism driving the seasonal changes in HPG axis responsiveness in birds compared to other vertebrates.

### 3.2. Seasonal Changes in Morphologic Characteristics of the Pineal Gland

The pineal gland of poultry is considered the most important light-sensing tissue, which exhibits a close association with the activity of the gonadal axis [45]. Studies have demonstrated that the pineal gland is an important neuroendocrine organ regulating reproduction by converting light-induced neural activity into endocrine hormones [46]. The pineal gland is thus defined as a photo-neuroendocrine converter and forms an essential part of the organism. It provides information about the photoperiod, thereby connecting the outside environments with the internal normal biochemical signaling and physiological needs of the body [47]. Typically, the size and activity of the pineal gland, which are related to the reproductive physiological state of animals, depend on environmental stimuli such as light stimulation. Seasonal breeding poultry mostly have a well-developed pineal gland [47]. Nonetheless, we observed a strange phenomenon in our study. We found that the weight and volume of the pineal gland in female geese showed seasonal changes contrary to gonadal activity, in accordance with similar results of Singh et al. (2007) in *Perdicula asiatica* [48]. Similar to most seasonal breeding mammalian species, there is an inverse relationship between the pineal gland and gonadal activity in some species of birds. Moreover, pineal functional modifications are also reflected in seasonal changes in structure. The histological structure of the goose pineal gland is similar to that observed by Wight et al. [45]. Pineal microvessels can offer vascular support for circannual periodic changes in the metabolic activity of the pineal tissue [49]. During the peak egg-laying period, along with the prolongation of the photoperiod came a decrease in the number of blood vessels around the connective tissue capsule of the pineal gland, resulting in a reduced secretory capacity. Therefore, the anatomical characteristics of the pineal gland under different photoperiods were closely related to the neuroendocrine regulation of seasonal reproduction in female geese.

### 3.3. Seasonal Variation in the Ultrastructure of Pineal Cells

The ultrastructure and functional changes in pineal cells also play an important role in regulating animal reproduction. In the experimental results of this study, the changes in the volume of the cell nucleus and the heterochromatin density might have been affected by the seasonal photoperiodic changes. As expected, on the basis of a previous study by Lee et al., there was a lower capacity of the pineal gland to secrete indoleamine hormones under a longer photoperiod, which resulted in promoting gonadal development [14]. Moreover, we observed through electron microscopy that the number of Golgi apparatus complexes increased slightly, gradually accumulating numerous vesicles at the AE. These characteristic morphological changes in the cell were suggestive of active secretory functions of the pineal gland, and the phenomena described here were consistent with the results reported by Frink et al. (1978) and McNulty et al. (1980) [50,51]. In addition, the current study confirmed that 5-HT is a fundamental component of melatonin biosynthesis [47]. Then, Aanat converts 5-HT into *N*-acetylserotonin, which is converted into melatonin by hydroxyindole-*O*-methyltransferase [52]. As such, we can only provide an educated guess here that the pineal cells might store a large amount of 5-hydroxytryptamine in order to synthesize melatonin under a short photoperiod of the AE. Meanwhile, glycogen was able to provide enough energy only for the physiological activities and biosynthesis of the cell [53]. More glycogen of pineal cells is needed in order to meet the consumption of melatonin synthesis. From this perspective, the ultrastructural changes in the pineal cells are an important indication of the seasonal physiological state of material synthesis and secretion.

### 3.4. Seasonal Characteristics of Melatonin Receptor Subtypes

Although the pineal gland is very important for seasonal breeding animals, melatonin needs to act on the hypothalamus–pituitary–gonadal axis by binding to the corresponding receptors [8]. Previous studies have confirmed that melatonin is intimately involved in regulation of the circadian rhythm, light signal transmission, and seasonality of the annual reproductive cycle [14,54]. In the present study, several notable findings were observed. For female geese, distribution analysis showed that the expression of three receptor genes had different seasonal changes in the pineal gland, hypothalamus, pituitary gland, and ovary. This is in contrast to the situation in mammals and fish, whereby *Mel-1c* was not detected in Atlantic salmon and mice, indicating adaptive evolution of organisms under varying environments. In addition, studies in seasonal breeding animals have found periodic changes in the expression levels of melatonin receptor genes concomitant with the photoperiod, and most of these have been reported in gonads (alongside some studies in brain tissue) [55]. Melatonin has been shown to act directly on the gonadal tissues [56,57,58]. In addition, the hypothalamus releases gonadotropin in response to melatonin, which indirectly inhibits its secretion [59]. In our study, the wide and specific tissue distribution of *Mtnrs* implied a series of processes regulated by melatonin. Alternatively, the three subtypes of *Mtnrs* (*Mel-1a*, *Mel-1b*, and *Mel-1c*) might have displayed a divergent distribution due to functionalization. However, unlike other nonseasonal breeding animals, changes in the expression levels of *Mtnrs* were present during the periodic changes, revealing the seasonal characteristics of melatonin in regulating reproduction.

### 3.5. Seasonal Effects of Melatonin on the Reproductive Endocrine System

Some studies on the control of sexual maturation in birds have explored the relationships between the seasonal photoperiod and the reproductive endocrine system [20]. Lee et al. observed an oscillation of melatonin levels with the natural pattern of seasonal photoperiod conditions [14]. In seasonal breeding animals, melatonin secretion fluctuates with photoperiod changes to regulate reproductive endocrine function. In addition, the ability of melatonin to regulate the reproduction of seasonal breeding animals has been proven several times in previous studies [60,61]. According to some studies, melatonin suppressed GnRH gene expression in GT1–7 cells and suppressed GnRH secretion by about 45%, indicating that melatonin could regulate GnRH neurons [62,63]. In the present study, we systematically assessed whether the rhythm of melatonin secretion could be correlated with seasonal changes in the neuroendocrine system at all levels of the HPG axis. Numerous studies have substantiated the crucial involvement of Gths, E2, and P4 in the developmental processes of gonads [64]. Among them, E2 assumes a pivotal role in various physiological functions encompassing growth, development, and reproduction [65]. Notably, it is plausible that the release of gonadal reserves of E2 could be triggered by GnRH [66]. In the present study, a lower level of melatonin regulated secretions of GnRH to stimulate secretions of LH, PRL, E2, and P4, along with gonadal development and reproductive activities, under a shorter photoperiod at the SE. Interestingly, the secretion of melatonin was inhibited by exposure to longer photoperiods, while the lowest amount of serum GnRH content was observed at the SS. There are results from the literature that also attest to this phenomenon. For instance, the findings by Yang et al. showed a close association of the reproductive hormones and their corresponding transcript expressions with the reproductive behaviors observed in geese [38]. Furthermore, some studies have observed that a high concentration of PRL could inhibit the secretion of pituitary gonadal hormone in the late laying period, leading to the interruption of reproductive behavior [11,67]. Afterward, the increase in melatonin content would further inhibit the activity of gonads with the increased daylength at the AE, bringing on the ceased-laying period for female geese. This suggests that the ocular melatonin signal is directly or indirectly related to reproduction. Further studies are needed to confirm the regulation methods underlying the reproductive state, but these findings suggest that, in female geese, melatonin can regulate seasonal reproduction upon receiving seasonal photoperiodic information.

## 4. Materials and Methods

### 4.1. Animals and Tissue Sampling

This study was performed on 760-day-old female Yangzhou geese raised according to common farming practices by Yangzhou Tiange Goose Industry Development Co., Ltd., Yangzhou, Jiangsu Province, China (32°64′ N, 119°34′ E). The geese lived in natural environmental conditions regarding the photoperiod. All animal experiments were approved by the Ethics Committee on Animal Experiments of Yangzhou University (Approval Code: 121-2018., Government of Jiangsu Province, China). These experiments did not involve the utilization of any rare or endangered animals. The protocols and methods were in accordance with the Regulations for the Administration of Affairs Concerning Experimental Animals, approved by the State Council of the People’s Republic of China.

The female Yangzhou geese selected for the experiments were hatched on 19 February 2018, 23 May 2018, 24 August 2018, and 22 November 2018. All Yangzhou geese (42 females and 7 males) were fed the same diet throughout the duration of the study (Appendix A), which combined coarse and concentrated material. Feed and water were provided ad libitum. The animals were tested at the spring equinox (SE), summer solstice (SS), autumn equinox (AE), and winter solstice (WS) in 2020, representing different seasonal patterns. The egg-laying rates of 42 female Yangzhou geese were recorded each day, and the photoperiod changes were also continuously monitored throughout the year. In each sampling season, six female geese were selected, and differences in the individual weight of the experimental animals in different seasons were within 3% of the mean weight (*N* = 6 per point). After blood sampling, serum was acquired via the centrifugation of blood samples at 3000× *g* for 15 min. Subsequently, the geese were sacrificed by anesthetizing them with sodium pentobarbital, before dissecting them to collect the pineal gland, hypothalamus, pituitary gland, and ovary, which were frozen in liquid nitrogen and stored at −80 °C until RNA extraction. Lastly, the pineal gland and ovary from the remaining six geese were collected and further subjected to histological observation.

### 4.2. Reproductive Performance and Histological Observations

The daily photoperiod is described as the change in day length, recorded throughout the year. In addition, the egg-laying rates of the geese were calculated on a daily basis as follows: egg-laying rate (%) = (number of eggs/total geese) × 100%. After MS-222 anesthesia, the sexually mature female geese (*N* = 6) were dissected to obtain the intact pineal gland and gonadal tissues in the different seasons, and the weight and gonadosomatic index (GSI) were measured. The latter was determined using the following formula: GSI (%) = [gonad weight (g)/total body weight (g)] × 100%. Next, samples from the pineal gland were fixed with 4% paraformaldehyde for 24 h at room temperature, dehydrated with graded ethanol, and embedded in paraffin. Sections of paraffin-embedded tissue (4 μm thick) were prepared for hematoxylin and eosin staining. The photographs were scanned using a Nanozoomer scanner (Hamamatsu, Sydney, Australia), and an image analysis system (Image-Pro Plus, Media Cybernetics, Rockville, MD, USA) was used to calculate the diameter of pineal gland follicles and the height of the pineal gland follicular cells for each sample.

For immunohistochemistry, the slides were dewaxed three times in xylene for 15 min each, and then heated with a microwave oven for antigen retrieval in Tris-ethylenediaminetetraacetic acid (EDTA) buffer (pH 9.0) at 95 °C for 20 min. The endogenous peroxidase activity was quenched with 3% H_2_O_2_ for 10 min. Then, the slides were incubated with the primary antibody anti-Mtnr (Mtnr, 1:1000, ab87639; Abcam, Cambridge, UK) at 4 °C overnight. The WB experiments were performed to verify the specificity of the Mtnr antibody (Appendix A). Slides were washed three times in phosphate-buffered saline (PBS) for 5 min each and incubated with the secondary antibody rabbit anti-mouse IgG (ab6728; Abcam, Cambridge, UK) for 30 min, followed by three additional washes in PBS and staining with 3,3′-diaminobenzidine (DAB, 1:20 dilution) for 10 min at room temperature. Lastly, after counterstaining with hematoxylin, samples were dehydrated in a graded ethyl alcohol series (70%, 90% and 100%), and cover slips were placed on. Bright-field and fluorescence images (GFP filter) of the stained sections were captured using an EVOS FLc imaging system (Thermo Fisher Scientific, Waltham, MA, USA). Image-Pro Plus (Image-Pro Plus, v6.0) was used to evaluate the mean integrated optical density (IOD) of the immunohistochemical results.

### 4.3. Semithin Sections and Transmission Electron Microscopy

The pineal glands were placed in electron microscopy liquid fixative (0.1% paraformaldehyde in 0.1 M sodium cacodylate) for TEM section preparation. The tissues were removed after a minimum of 12 h of fixation. Slides of the pineal gland were dehydrated in an ascending ethanol series, and then transferred to propylene oxide and embedded in Epon (Sigma Aldrich, St. Louis, MO, USA). Semithin sections, stained with 1% toluidine blue (Reanal, Budapest, Hungary), were used for orientation; then, ultrathin sections were cut with a glass knife, mounted on grids, and allowed to dry. Uranyl acetate and lead citrate (Ted Pella Inc., Redding, CA, USA) were used as post-embedding staining. The slide samples were studied using a JEOL 1010 transmission electron microscope (JEOL, Peabody, MA, USA) operating at 80 kV.

### 4.4. RNA Extraction and Quantitative RT-PCR

The following experiments were conducted to measure gene expression: “experiment 1”—quantification of seasonal *Aanat* expression across different tissues; “experiment 2”—tissue distribution of *Mtnrs* expression in the four seasons; “experiment 3”—seasonal expression of genes relevant to the hypothalamus–pituitary–gonadal (HPG) axis.

Total RNA was extracted using Trizol (Invitrogen, San Diego, CA, USA) according to the instructions of the manufacturer. The concentration and purity of RNA samples were determined using a NanoDrop™ ND-1000 spectrophotometer (NanoDrop Technologies, Wilmington, DE, USA), and the RNA integrity was verified by electrophoresis on 1% agarose gel. One microgram of total RNA was reverse-transcribed with a FastQuant RT Kit (with gDNase) (Takara Biotechnology Co., Ltd., Dalian, China) following the supplier’s protocol. The RT reaction was performed as follows: the reaction mixture was incubated for 15 min at 42 °C to synthesize cDNA, followed by 3 min at 95 °C. The resulting cDNA samples were diluted fivefold and stored at −20 °C until further analysis. The primers for qRT-PCR (Table 2) were designed using Primer software (Version 5.0, Primer, Kingston, ON, Canada) and were synthesized by TSINGKE Biological Technology (Nanjing, China). Glyceraldehyde-3-phosphate dehydrogenase (*GAPDH*) was used as the reference control gene to normalize target gene expression. Then, qRT-PCR was carried out using SYBR Green Master Mix (ABclonal, Wuhan, China), and data were analyzed in Quant Studio 5 (Applied Biosystems, Thermo Fisher Scientific, Waltham, MA, USA). The following qRT-PCR thermal cycling program was employed: 95 °C for 5 min, and then 41 cycles of 95 °C for 20 s and 60 °C for 30 s, which included data acquisition. Relative expression levels were determined using the 2^−ΔΔCt^ method, where the ΔCt value was derived as a function of the difference between the Ct value of each tested gene and that of the reference gene. Each reaction was performed in triplicate, and the data were the average of three independent experiments.

### 4.5. Measurement of Serum Concentrations of Hormones

For the estimation of hormones, blood was centrifuged, and serum was collected for the analysis of melatonin (MT), gonadotropin-releasing hormone (GnRH), follicle-stimulating hormone (FSH), luteinizing hormone (LH), prolactin (PRL), estradiol (E2), and progesterone (P4) levels using commercially available enzyme-linked immunosorbent assay (ELISA) kits (Wuhan EIAab Science Co. Ltd., Wuhan, Hubei, China). Operations were carried out according to the manufacturer’s protocols. The sensitivities of the ELISAs were 10.0 pg/mL, 10.0 pg/mL, 0.56 mIU/mL, 0.19 mIU/mL, 0.195 pg/mL, 0.056 ng/mL, and 0.15 ng/mL, respectively. There was no cross-reaction with other structural analogues, and all the intra-assay and inter-assay coefficients of variation (CVs) for each hormonal assay were less than 10% and 15%, respectively. Lastly, the absorbance of 450 nm was read for each well using a microplate reader, Model 680 (BioRad, Hercules, CA, USA). The ELISA standard curves were analyzed using a four-parameter logistic equation:y = A2 + (A1 − A2)/(1 + (x/x0)^p^).

Standard curves with *R*^2^ ≥ 0.98 were accepted.

### 4.6. Statistical Analysis

Results were expressed as the mean ± *SEM* (standard error of the mean). All data were checked for normality using the Kolmogorov–Smirnov test of normality before being analyzed. Depending on the design of the experiment, data were analyzed using a one-way (experiments 1 and 3, as well as hormone and histological experiments) or two-way ANOVA (experiment 2), followed by Tukey’s post hoc test. Significance was set at the 0.05 level (*p* < 0.05). Statistical analyses were performed using SPSS 20.0 software package (IBM Corp, Armonk, NY, USA).

## 5. Conclusions

Our study demonstrates that Yangzhou geese are long-day breeders. The seasonal regulation of the reproductive axis plays an important role in the reproductive function throughout the year. In addition, the seasonal reproductive rhythms suggest that the change in natural photoperiod signal is a significant regulator. As a photoperiod transducer, the seasonal changes in pineal melatonin secretion drive the alternation between the laying period (LP) and ceased-laying period (CP). Lastly, our study provides basic information on photoperiodic regulation of seasonal reproduction, along with clues of what underlies the regulatory relationship between pineal melatonin and the reproductive endocrine system in seasonal breeding animals.

## Figures and Tables

**Figure 1 ijms-24-11998-f001:**
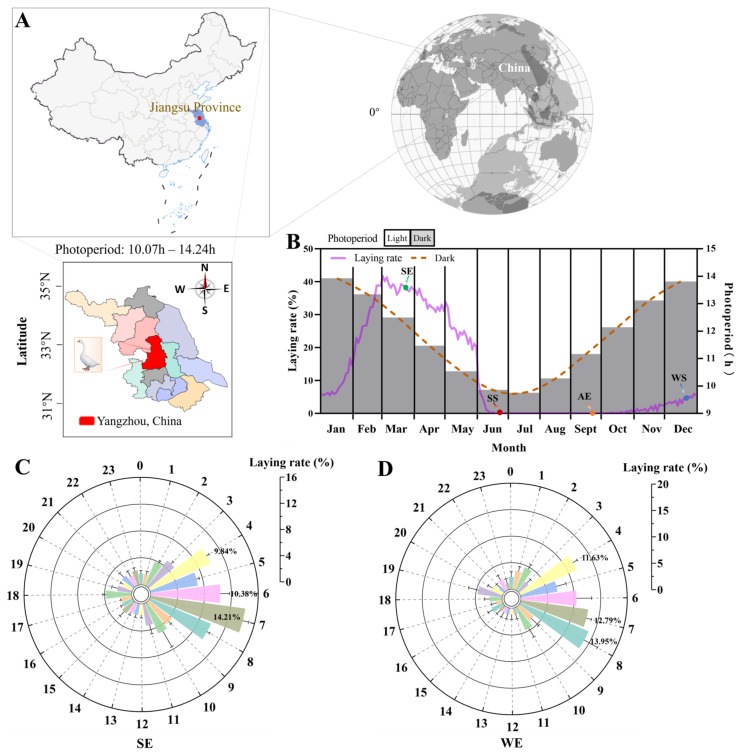
The seasonal changes in the reproductive activity of female Yangzhou geese under the natural photoperiod. (**A**) Location map of the experimental site, labeled as the red region. Latitude was taken from Google Earth (www.googleearth.com, accessed on 5 January 2021). Colors indicate the cities: red represents the area of sample collection (Yangzhou, China). (**B**) Annual changes in the natural photoperiod (bar chart) and laying rate (solid line) are shown during the entire experimental period. SE, Spring equinox (20 March 2020); SS, summer solstice (21 June 2020); AE, autumn equinox (22 September 2020); WS, winter solstice (21 December 2020). (**C**,**D**) The observations of the egg-laying time at the SE and WE. Outer circles represent the 24 h day, and vertical coordinates indicate the laying rate. Colors correspond to time points.

**Figure 2 ijms-24-11998-f002:**
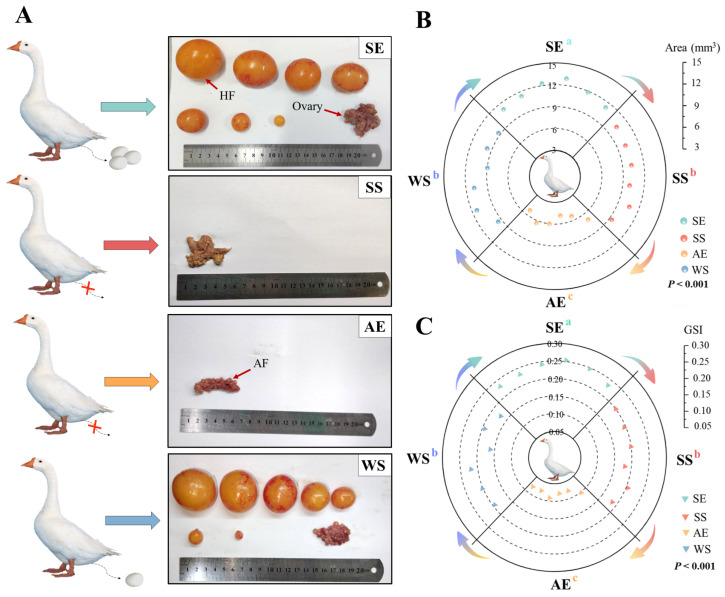
Effect of seasonal photoperiods on the gonad development. (**A**) Gross anatomical changes in the ovarian and follicular development at the SE, SS, AE, and WS. HF, hierarchical follicles; AF, atretic follicles. Cross symbol (×) indicates stop laying eggs. (**B**) Ovarian area (SE, *SEM =* 0.288; SS, *SEM* = 0.179; AE, *SEM* = 0.317; WS, *SEM* = 0.382). (**C**) Gonad index (GSI) (SE, *SEM =* 0.006; SS, *SEM* = 0.004; AE, *SEM* = 0.005; WS, *SEM* = 0.007). Different letters indicate statistically significant differences among groups (*p* < 0.05), analyzed using one-way ANOVA followed by Tukey’s multiple comparisons test (*N* = 6 per point). ^a,b,c^ Different letters represent significant differences.

**Figure 3 ijms-24-11998-f003:**
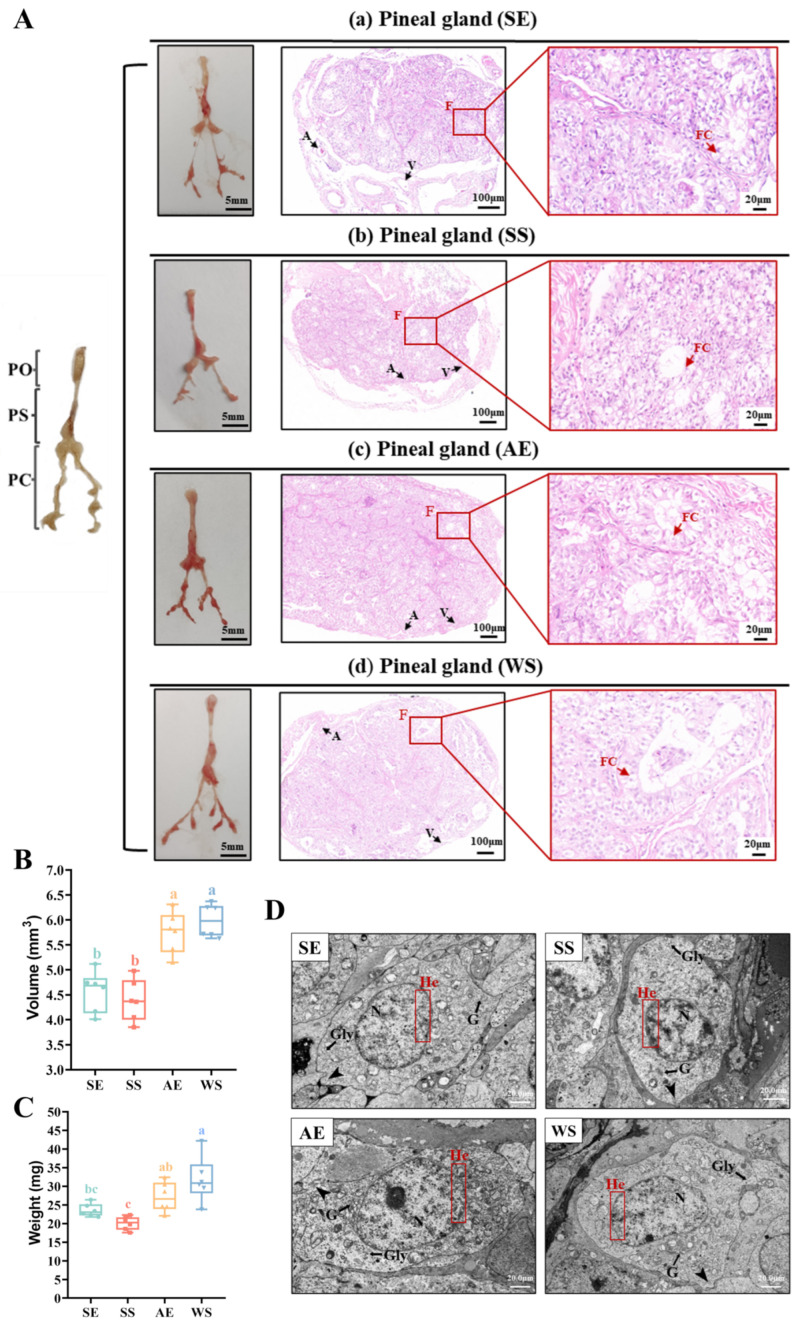
The morphological characteristics and seasonal changes in the pineal gland under different photoperiods. (**A**) The anatomical observation of pineal gland development (PO = pineal organ, PS = pineal stalk, and PC = pineal choroid). Hematoxylin–eosin (HE) staining was conducted to observe the pineal gland. A, artery; V, vein; F, follicle-like structure; FC, follicular cell. (**B**,**C**) Seasonal changes in the volume and weight of the pineal gland in female Yangzhou geese. The data are presented as the mean ± *SEM*. Different letters indicate statistically significant differences among groups (*p* < 0.05), analyzed by one-way ANOVA followed with Tukey’s multiple comparisons test. (**D**) Microstructure of pineal gland cells examined under transmission electron microscopy (TEM). The nuclei are heterochromatic, and the DNA is tightly packed. High-magnification views of heterochromatin around the nuclear envelope are shown on the right. Arrows indicate astrocytic processes. Arrowheads indicate Golgi apparatus (G) and glycogen particles (Gly). N, nucleus; He, heterochromatin.

**Figure 4 ijms-24-11998-f004:**
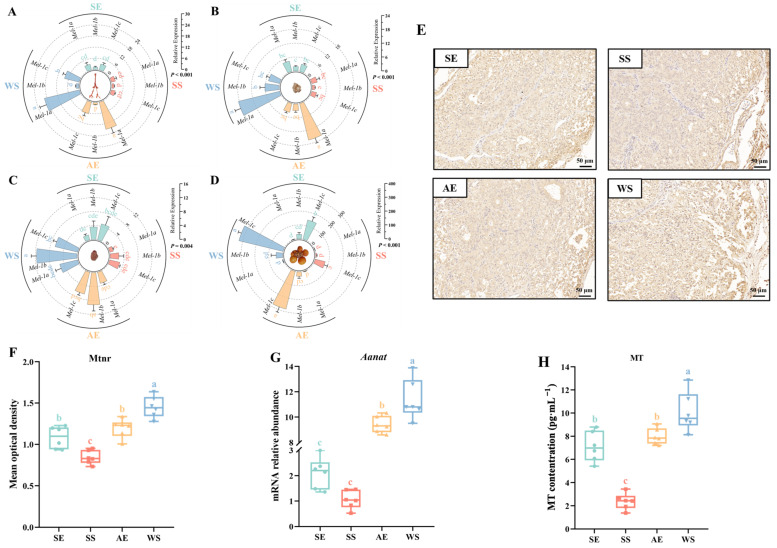
The expression of melatonin biofunction-related genes and the endogenous melatonin level of female Yangzhou geese in the pineal gland and HPG axis (*N* = 6 per point). Relative expression of *Mtnrs* in the pineal gland (**A**), hypothalamus (**B**), pituitary gland (**C**), and ovary (**D**). mRNA levels are normalized to *GAPDH*. The vertical coordinates indicate the mRNA relative expression. The data are presented as the mean ± *SEM*. Different letters indicate statistically significant differences among groups (*p* < 0.05), analyzed using two−way ANOVA followed by Tukey’s multiple comparisons test. (**E**) Representative images of immunohistochemistry staining. (**F**) Average optical density of pineal gland tissue IHC detection. (**G**) Expression changes of *Aanat* mRNA in the pineal gland. (**H**) Melatonin (MT) content in the serum. Statistically significant differences are denoted by different letters (*p* < 0.05, one-way ANOVA with Tukey’s test).

**Figure 5 ijms-24-11998-f005:**
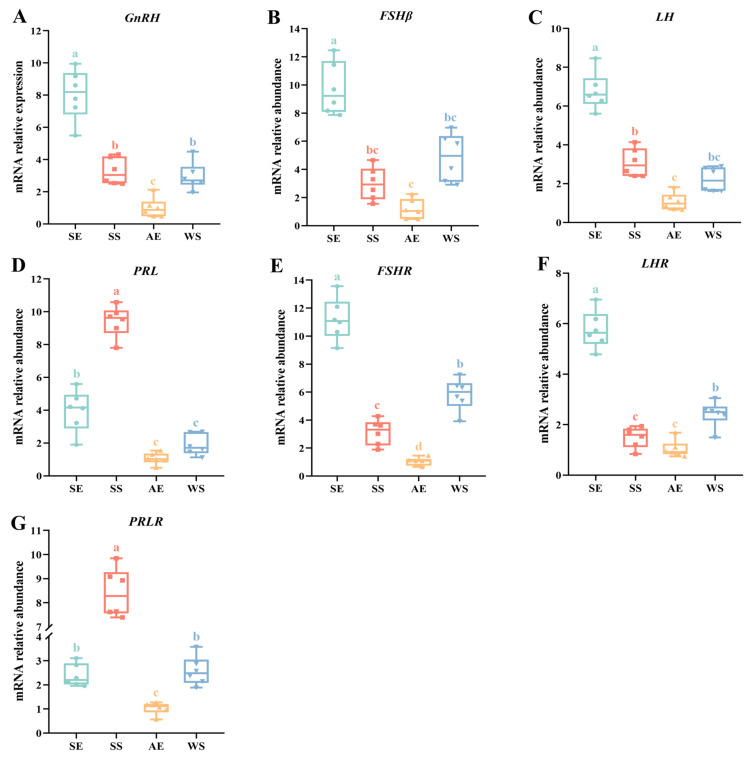
The relative mRNA expression of reproduction-related genes in the pineal gland, hypothalamus, pituitary gland, and ovary (*N* = 6 per point). (**A**) Expression of reproduction-related gene (*GnRH*) mRNAs in the hypothalamus. (**B**–**D**) Expression of reproduction-related gene (*FSH*, *LH*, and *PRL*) mRNAs in the pituitary gland. (**E**–**G**) Expression of reproduction-related gene (*FSHR*, *LHR*, and *PRLR*) mRNAs in the ovary. mRNA levels are normalized to *GAPDH*. The data are presented as the mean ± *SEM*. Different letters indicate statistically significant differences among groups (*p* < 0.05), analyzed using one-way ANOVA followed by Tukey’s multiple comparisons test.

**Figure 6 ijms-24-11998-f006:**
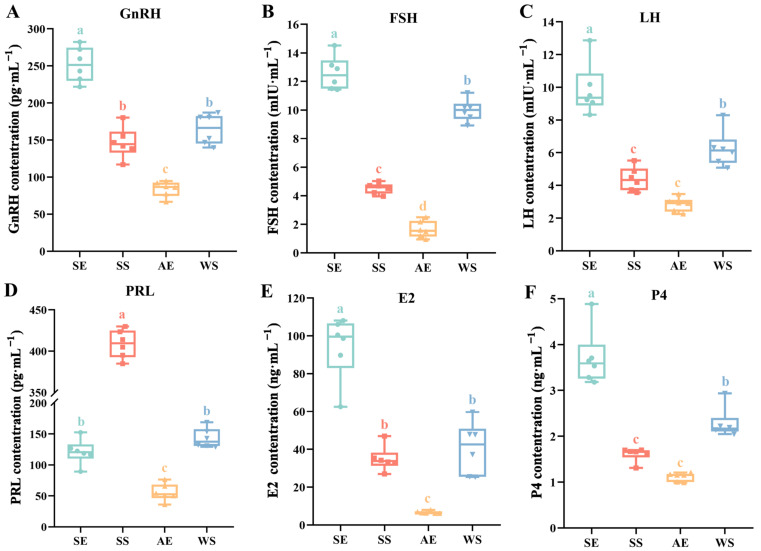
Effects of natural photoperiod exposure on serum parameters of female Yangzhou geese over the course of different seasons (*N* = 6 per point). (**A**) GnRH, gonadotropin-releasing hormone; (**B**) FSH, follicle-stimulating hormone; (**C**) LH, luteinizing hormone; (**D**) PRL, prolactin; (**E**) E2, estradiol; (**F**) P4, progesterone. The data are presented as the mean ± *SEM*. Different letters indicate statistically significant differences among groups (*p* < 0.05), analyzed using one-way ANOVA followed by Tukey’s multiple comparisons test.

**Figure 7 ijms-24-11998-f007:**
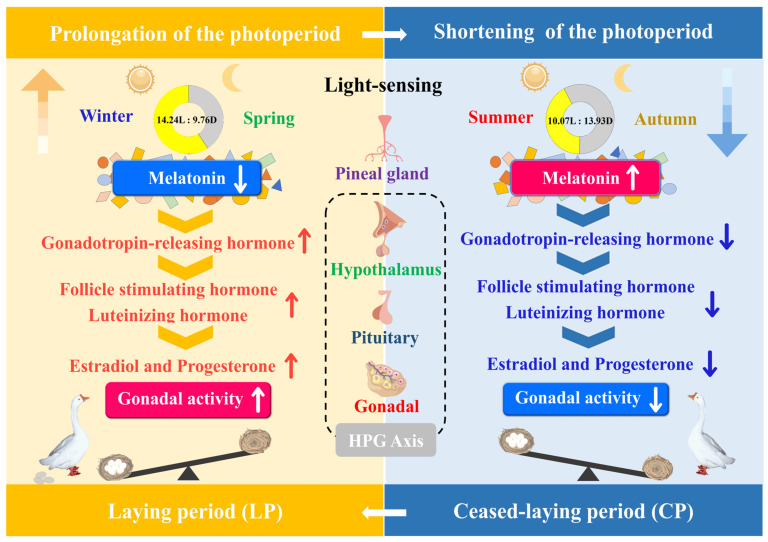
Schematic summarizing the interaction mechanism between the annual variation in photoperiod and seasonal reproduction. Red upward arrow and blue downward arrow denote up- and downregulation, respectively.

**Table 1 ijms-24-11998-t001:** Structural characteristics of follicle-like structure in different seasons.

Variable	Season	Number	Long Diameter (μm)	Cell Height (μm)
Mean	SE	29.33 ^b^	70.10 ^b^	18.25 ^a^
	SS	26.16 ^b^	51.40 ^b^	12.26 ^b^
	AE	36.16 ^a^	66.93 ^b^	18.43 ^a^
	WS	35.83 ^a^	107.55 ^a^	19.96 ^a^
*Pooled SEM*		2.23	8.20	1.50
*F* (*df*)		9.822 (5)	16.871 (5)	10.218 (5)
*p*-value		0.000	0.000	0.000

Note: The number, long diameter, and cell height of follicle-like structures in the pineal gland are shown. Different letters indicate statistically significant differences among groups (*p* < 0.05), analyzed using one-way ANOVA followed by Tukey’s multiple comparisons test. *F* = *F* statistic; *p* = *p*-value for *F* statistic; *df* = degrees of freedom.

**Table 2 ijms-24-11998-t002:** The primers used in a quantitative RT-PCR assay for gene expression.

Gene Name	Accession Number	Primer Sequence (5′–3′)	PCR Product (bp)
*Aanat*	XM_048064549.1	F: GGTAAGCCCACGGTTCTGTT	125
		R: CTTCCCTCCGGGACAATTCC	
*Mel-1a*	XM_048078352.1	F: TCATGCACGTTTGCACAGTC	169
		R: GTGGTCTCAGTCTGGGGTTG	
*Mel-1b*	XM_013178069.2	F: CACGGTGGTGGACATCT	139
		R: CAGTGGGTATGGATACAAGG	
*Mel-1c*	XM_048075512.1	F: CAGATAAGTGGGTTCCTGATGGG	103
		R: ACCGAAGGCTGTGGCAGATGTAG	
*GnRH* ^a^	DQ023158	F: GAAGATCTTGGTCGGTGTCCTCCTGT	262
		R: AATCTCCTTTCTTCTGGCTTCTCCTTC	
*FSHβ*	XM_013177587.2	F: CACCAGTATCATCCGTTCAGC	153
		R: CAGTGCTATCAGTGTCACAGGTC	
*FSHR*	XM_013192471.2	F: TGCCAGGTCACGGATTAGAAC	165
		R: ATTCAGTGTTTTGTCTTTTCCAGT	
*LH* ^b^	DQ023159	F: GGTGTATCGCAGCCCTTTG	133
		R: TATCAGAGCCACGGGGAGG	
*LHR*	XM_048078485.1	F: CTCTGTGATAACTTGCGTAT	119
		R: AAGGCATGACTGTGGAT	
*PRL*	XM_013184821.2	F: CCTGAAGACAAGGAGCAAGC	222
		R: AGAATGAACCCGCCCAAC	
*PRLR*	XM_048051239.1	F: GCCTTTATCCTACCACCAGTTCC	175
		R: GATCCTCGCTGTCCTCTACCTCT	
*GAPDH*	XM_013199522.2	F: TCGGAGTCAACGGATTTGGC	175
		R: TTCTCAGCCTTGACTGTGCC	

^a^ Primers from Huang et al. (2008) [68]. ^b^ Primers from Zhang et al. (2013) [69].

## Data Availability

The data presented in this study are available upon request from the corresponding author.

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
