# Peer review of "The Photoperiod-Driven Cyclical Secretion of Pineal Melatonin Regulates Seasonal Reproduction in Geese (Anser cygnoides)"

_ijms, 2023, doi:10.3390/ijms241511998_

Round 1

Reviewer 1 Report

The abstract need to be improved. The sentence introducing the Mtnrs genes needs to be introduced by specifying that these are genes for melatonin receptors. Same for other genes like Anat, GnRH and GtHs. 

Data on laying rate and hormones in goose were previously reported by H.M. Yang, Y. Wang, Z.Y. Wang, X.X. Wang "Seasonal and photoperiodic regulation of reproductive hormones and related genes in Yangzhou geese". Poultry Science.

Statistical analysis are a major concern. While the initial number of animals per group was 42, only 6 animals were reported for the subsequent experiments. Furthermore, no standard error of the mean (SEM) or standard deviation (SD) values were reported in Figures 1B, 1C, 1D, 2B, and 2C. All p-values and F values for the ANOVA analysis need to be reported to ensure transparency and provide a comprehensive understanding of the results.

English Grammar needs improvement

Author Response

Response to Reviewer 1 Comments

Point 1: The abstract need to be improved.

The sentence introducing the Mtnrs genes needs to be introduced by specifying that these are genes for melatonin receptors. Same for other genes like Anat, GnRH and GtHs.

Response 1: We thank for your insightful and constructive comments. Those comments are all valuable and very helpful for revising and improving our paper, as well as the important guiding significance to our researches. We have studied comments carefully and have made correction which we sincerely hope to get your approval. The Abstract was modified to provide more relevant information related to genes. The relevant sentences were revised as follows:

Notably, tissue distribution analysis further revealed seasonal oscillations of melatonin receptors (Mtnrs) in pineal gland and hypothalamus-pituitary-gonadal (HPG) axis (Lines 21-23, page 1).

Furthermore, the upregulation of Aralkylamine N-Acetyltransferase (Aanat) was observed from the SS to AE, while concurrently resulting in a downregulation of gonadotrophin-releasing hormone (GnRH) and gonadotropins (GtHs) (Lines 24-26, page 1).

Once again, thank you very much for your comments and suggestions.

Point 2: Data on laying rate and hormones in goose were previously reported by H.M. Yang, Y. Wang, Z.Y. Wang, X.X. Wang "Seasonal and photoperiodic regulation of reproductive hormones and related genes in Yangzhou geese". Poultry Science.

Response 2: Thank you for a good question. Our deepest gratitude goes to you for your careful work and thoughtful suggestions that have helped improve this paper substantially. Of course, we understand your concern that a similar study has already been conducted by Yang et al. This is an interesting and useful study published in Poultry Science.

In response to your concern, we have carefully examined the existing literature and compared our research with the previously published work. While there might be some similarities in the objectives, our study offers several novel contributions and advancements to the field. We believe that our research provide additional insights and expand the existing literature in several important ways.

Specifically, we would like to highlight the following points that differentiate our research from the previous study.

  1. Novelty of the methodologies: Our research adopts a novelty methodology that has not been explored in previous studies. We explored a hypothesis that the pineal gland directly senses the changes in the length of light throughout the year to regulate the cyclical secretion of melatonin, and thereby controls the gonadal development and seasonal reproduction in geese. Our results contribute to a better understanding of the physiologic regulatory mechanism underlying seasonal reproduction of domestic goose.
  2. Comprehensive trials and analysis: We have collected a more comprehensive dataset that encompasses a broader range of variables and a larger sample size compared to the previous study. Moreover, we have conducted additional experiments and analysis that were not performed in the previous study. We conducted an examination of the histological characteristics of the pineal gland and ovary. On the other hand, a stratified design was used for the experiments, the same day-age (760-day-old) of female geese from different seasons were performed in our study. Unlike previous studies, this experiment effectively mitigated the influence of age on the geese in different seasons, thereby eliminating the confounding effect of growth and development. Thus, these novel methodologies and analytical approaches provide unique insights into the research problem and enhance the generalizability of our findings.
  3. A broader implication: Our study provided basic information on photoperiodic regulation of seasonal reproductive and used as a clue that provided the inspiration on the regulation between pineal melatonin and reproductive endocrine in seasonal breeding animals. As a result, these data deepened the understanding of the photoperiodic regulation of seasonal reproduction and provided a valuable theoretical basis for improving the egg production in geese.

We understand the importance of avoiding redundancy and duplicative research efforts in the scientific community. Therefore, we have taken great care to ensure that our research adds value and complements the existing body of knowledge. Meanwhile, we have taken care to cite and discuss the relevant prior work in our manuscript to highlight the existing knowledge and contextualize our contributions appropriately (Lines 299-301 ,page 11).

Finally, we extremely grateful your valuable feedback and assure that we have addressed your concern by highlighting the distinct contributions of this research. We believe that our manuscript merits publication in “International Journal of Molecular Sciences” due to its distinct contributions and significance to the field.

Point 3: Statistical analysis are a major concern. While the initial number of animals per group was 42, only 6 animals were reported for the subsequent experiments. Furthermore, no standard error of the mean (SEM) or standard deviation (SD) values were reported in Figures 1B, 1C, 1D, 2B, and 2C. All p-values and F values for the ANOVA analysis need to be reported to ensure transparency and provide a comprehensive understanding of the results.

Response 3: We are extremely grateful to reviewer for pointing out this problem. We are sorry for the unclear expression. For clarity, we have made some modifications in Materials and Methods section as follows: “All Yangzhou geese (42♀+7♂) were fed the same diet throughout the duration of the study, which was combined with coarse and concentrated material. Feed and water were provided ad libitum. They were tested at the spring equinox (SE), summer solstice (SS), autumn equinox (AE), and winter solstice (WS) in 2020, representing different season pattern. The egg-laying rate of 42 female Yangzhou geese were record each day, and the photoperiods change were also continuously monitored throughout the year. In each sampling season, six female geese were selected and differences in the individual weight of the experimental animals in the different season were within 3% of mean weight (N = 6 for per point) (Lines 441-449, page 14).”

Moreover, we sincerely thank the reviewer for thoroughly examining our manuscript and providing very helpful comments to guide our revision. According to the Reviewer’s comments, we now quantified the data and provide the standard error of the mean (SEM) in Figures 1B, 1C, 1D, 2B, and 2C. Meanwhile, all P-values and F-values for the ANOVA analysis were reported correctly in the revised version of the manuscript.

Point 4: English Grammar needs improvement

Response 4: Thank you for your careful review. We are very sorry for the mistakes in this manuscript and inconvenience they caused in your reading. The manuscript has been thoroughly revised and rewritten by a native English speaker, so we hope it can meet the journal’s standard.

Once again, thank you very much for your comments and suggestions.

Reviewer 2 Report

In this paper, Bao and colleagues analyzed the effect of photoperiod on reproductive activity in goose (Anser cygnoides). They collected samples at the Spring Equinox (SE), Summer Solstice (SS), Autumn 16 Equinox (AE), and Winter Solstice (WS). They observed an increase in the weight and volume of the pineal gland, as well as of its secretory activity, with a shortening photoperiod.  In addition, they found seasonal oscillations in the expression of melatonin receptors and sexual hormones. 

Thus, they concluded that melatonin is secreted during specific seasons, and it regulates the neuroendocrine function of reproduction. 

Theoretically, the paper is interesting and well conducted; however, I have a big concern about the used experimental animal, indeed Anser cygnoides has been evaluated as “…vulnerable because it is suspected to be undergoing a rapid population decline owing to poor breeding success in recent years as a result of drought and considerable pressure from habitat loss, particularly owing to agricultural development, as well as unsustainable levels of hunting.” (https://www.iucnredlist.org); for this, the authors should extensively and convincingly justify their choice. 

Apart from this, here are other comments:

- In the abstract, the used animal model should be specified;

- It is known that Kiss neurons regulate GnRH secretion; therefore, the authors should cite this in the introduction (line 57) and also analyze whether photoperiod has an effect on Kiss;

- Why just female animals were considered?

- In lines 64-65, the authors assess that “As compared to other vertebrates, birds may have highly sophisticated regulation mechanisms”, so they should better clarify which are the differences between birds and other vertebrates;

- Sentence in lines 93-95 is confusing and should be rephrased;

- Histology images in Fig. 3A and 4E are too small and should be enlarged; the same for the labeling of all the graphs in Figs. 4, 5, and 6;

- Line 402: probably, the animal age should be modified;

- The used Mtnr antibody has been validated just for Western blot analysis in human samples; thus, the authors should provide evidence for its specificity in goose;

- Line 488: DDC(t) and DCt should be ΔΔC(t) and ΔCt;

- Do the authors have information on estradiol concentration?

Author Response

Response to Reviewer 2 Comments

Point 1: In this paper, Bao and colleagues analyzed the effect of photoperiod on reproductive activity in goose (Anser cygnoides). They collected samples at the Spring Equinox (SE), Summer Solstice (SS), Autumn 16 Equinox (AE), and Winter Solstice (WS). They observed an increase in the weight and volume of the pineal gland, as well as of its secretory activity, with a shortening photoperiod. In addition, they found seasonal oscillations in the expression of melatonin receptors and sexual hormones. Thus, they concluded that melatonin is secreted during specific seasons, and it regulates the neuroendocrine function of reproduction.

Theoretically, the paper is interesting and well conducted; however, I have a big concern about the used experimental animal, indeed Anser cygnoides has been evaluated as “…vulnerable because it is suspected to be undergoing a rapid population decline owing to poor breeding success in recent years as a result of drought and considerable pressure from habitat loss, particularly owing to agricultural development, as well as unsustainable levels of hunting.” (https://www.iucnredlist.org); for this, the authors should extensively and convincingly justify their choice.

Response 1: Thanks very much for taking your time to review this manuscript. We really appreciate the reviewer’s positive evaluation of our work. Those comments are all valuable and very helpful for revising and improving our paper. We tried our best to improve the manuscript and made some changes in the manuscript. These changes will not influence the content and framework of the paper.

Moreover, we perfectly understand the reviewer's concern about the used experimental animal in this study. Conservation of the genetic resources of endangered animals is crucial for future generations.

We have now provided detailed information on animals used as follow.

Herein, Yangzhou geese (Anser cygnoides) were selected at the Spring Equinox (SE), Summer Solstice (SS), Autumn Equinox (AE), and Winter Solstice (WS), and the regulation of seasonal reproduction by light-driven cyclical secretion of pineal melatonin was investigated (Lines 16-18, page 1). And then, the text geese are one of Chinese indigenous domestic goose breeds, mainly located in the northern of Jiangsu Province in China (Lines 121-122, page 3). As per the latest (livestock and poultry) census in China, Yangzhou goose (Anser cygnoides) has a relatively large group (http://www.stats.gov.cn/sj/). Therefore, this experiment did not involve the utilization of any rare or endangered animals (Lines 435-436, page 14).

Once again, we appreciate for your warm work earnestly, and hope that the correction will meet with approval.

Point 2: In the abstract, the used animal model should be specified.

Response 2: Thank you for your constructive comments on my manuscript. We have carefully considered the suggestion of reviewer and made the Abstract section much more detailed. In our study, Yangzhou geese (Anser cygnoides) was used as the study model.

The manuscript has been revised according to the comments as follows:“ Herein, Yangzhou geese (Anser cygnoides) were selected at the Spring Equinox (SE), Summer Solstice (SS), Autumn Equinox (AE), and Winter Solstice (WS), and the regulation of seasonal reproduction by light-driven cyclical secretion of pineal melatonin was investigated (Lines 16-18, page 1).”

Point 3: It is known that Kiss neurons regulate GnRH secretion; therefore, the authors should cite this in the introduction (line 57) and also analyze whether photoperiod has an effect on Kiss.

Response 3: We truly appreciate the reviewer’s professional suggestions. We have updated the text to include recent studies regarding to Kiss neurons and GnRH secretion in the revised manuscript as follows: “Numerous studies have found that reproductive function is coordinated by kisspeptin (Kiss) neurons and GnRH. The hypothalamus integrates signals from external stimuli to generate Kiss, a ligand of the G-protein coupled receptor GPR54, which in turn activates GnRH neurons to initiate the production of GnRH. In avian species literature, however, photoperiodic control of kisspeptin is relatively scarce and controversial (Lines 59-64, page 2).”

Moreover, one study reported that the KiSS/KiSSR system is completely absent, indicating that KiSS signaling is not required in birds (Reference: Kim, D. K., Cho, E. B., Moon, M. J., Park, S., Hwang, J. I., Do Rego, J. L., Vaudry, H., Seong, J. Y. Molecular Coevolution of Neuropeptides Gonadotropin-Releasing Hormone and Kisspeptin with their Cognate G Protein-Coupled Receptors. Frontiers in neuroscience 2012, 6, 3. https://doi.org/10.3389/fnins.2012.00003.). This is a very interesting finding. Therefore, we will further explore the regulatory role of Kiss neurons in poultry reproductive function in future research. Meawhile, there are limited reports on the relationship between photoperiod and Kiss neuronal function. Another good idea that we will try to explore the effect of photoperiod on Kiss after this study.

Thank you again for your valuable suggestions.

Point 4: Why just female animals were considered?

Response 4: Thanks for the reviewer raising this very valuable question. In this study, female geese were used as the experimental object for the following reasons.

First, Seasonally breeding animals, especially females, characterized by physiological changes, growth, and development (Reference: Garcia C, Huffman M, Shimizu K. Seasonal and reproductive variation in body condition in captive female Japanese macaques (Macaca fuscata). Am J Primatol. 2010, 72, 277-286. doi:10.1002/ajp.20777)

And second, the laying performance of female goose is an important economical trait in goose production. In seasonally breeding animals, timing of reproduction in females can influence offspring development and survival (Reference: Siutz C, Millesi E. Effects of birth date and natal dispersal on faecal glucocorticoid concentrations in juvenile Common hamsters. Gen Comp Endocrinol. 2012, 178, 323-9. doi: 10.1016/j.ygcen.2012.06.009.). In addition, females animals in their reproductive years are more sensitive to seasonal variation. During different breeding seasons, the hormone levels and gonad development of female animals will undergo a suite of physiological change. They are also more obvious and easier to observe than that in ganders.

Finally, the sample collection and experimental design are also one of the important reasons for choosing female geese as research objects.

However, this does not mean that the role of male animals in reproduction is not important. Therefore, we fully agree with the reviewer's concern. In future studies, we will explore the role and influence of male animals in seasonal reproduction, so as to gain a more comprehensive understanding of the intrinsic molecular regulation mechanism of seasonal reproduction.

We now provide these details in the Introduction and Discussion (Lines 39-40, page 1; Lines 294-297, page 11).

Thanks again!

Point 5: In lines 64-65, the authors assess that “As compared to other vertebrates, birds may have highly sophisticated regulation mechanisms”, so they should better clarify which are the differences between birds and other vertebrates;

Response 5: We deeply appreciate the reviewer’s suggestion. According to the reviewer’s comment, we have added the detailed information in the revised manuscript as follows: “Several studies have found that the hypothalamic suprachiasmatic nucleus (SCN) serves as the primary photoreceptive system in most vertebrates, whereas birds possess a photoreceptive system that encompasses the pineal gland and SCN. Notably, the pineal gland assumes a crucial role in modulating the annual rhythm through light signal transmission (Lines 72-76, page 2).” Moreover, the reference was added accordingly.

Point 6: Sentence in lines 93-95 is confusing and should be rephrased.

Response 6: Thanks for pointing out the problem. We apologize for the confusion generated by the previous version of the manuscript and sincerely hope that our logic is now easier to follow with this new version it.

We have modified the sentence to make it clearer: “The Mel-1a and Mel-1b are present in humans and other mammals, while Mel-1c as an additional melatonin receptor subtype has been identified in fish, amphibian and bird (Lines 102-104, page 3).”

Point 7: Histology images in Fig. 3A and 4E are too small and should be enlarged; the same for the labeling of all the graphs in Figs. 4, 5, and 6 .;

Response 7: We thank the reviewer for the insightful and very constructive comments, which were helpful in revising and strengthening this manuscript. The manuscript has been revised according to the comments. Histology images in Figure 3A and 4E are enlarged, and representative enlarged images are shown in Figure 3A (right panels). Moreover, we removed the same for the labeling of all the graphs in Figs. 4, 5, and 6.

Point 8: Line 402: probably, the animal age should be modified.

Response 8: The reviewer is right .We thank you for pointing out this problem. According to your advice, we amended the relevant part in manuscript. The animal age has been modified to “ 760-day-old” (Line 428, page 14).

Point 9: The used Mtnr antibody has been validated just for Western blot analysis in human samples; thus, the authors should provide evidence for its specificity in goose.

Response 9: We sincerely thank the reviewer for raising this professional question and helpful sugestions. We echo the sentiment of the reviewer. We have provided more detailed and important information in the Materials and Methods. The WB experiments were performed to verify the specificity of the Mtnr antibody (Figure S1) (Lines 473-474, page 15).Therefore, we believe that these data highlight the robustness and specificity of this antibody.

Point 10: Line 488: DDC(t) and DCt should be ΔΔC(t) and ΔCt. 第488行:DDC(t)和DCt应为ΔΔC(t)与ΔCt。

Response 10: We are extremely grateful to editor for pointing out this problem. We have modified this sentence as “Relative expression levels were determined using the 2–ΔΔC(t) method, where ΔCt value was derived based on the difference between the Ct value of each tested gene and that of the reference gene (Lines 515-517, page 16).”

Point 11: Do the authors have information on estradiol concentration.

Response 11: Thank you for your suggestions. We have provided more information on estradiol concentration. The related statement had also been revised in manuscript. we have now replaced “concent” with “concentration” (Lines 252 and 254, page 10). In line with this modification, Figure 6 have been revised accordingly.

In addition, more information about estradiol have been added in the revised manuscript as follows:” Numerous studies have substantiated the crucial involvement of Gths, E2, and P4 in the developmental processes of gonads. Among them, E2 assumes a pivotal role in various physiological functions encompassing growth, development, and reproduction. Notably, it is plausible that the release of gonadal reserves of E2 could been triggered by GnRH (Lines 406-410, page 14).”

Thank you again for the opportunity to revise our manuscript. Those comments are all valuable and very helpful for revising and improving our paper. We have tried our best to revise our manuscript according to the comments and given responses point by point. We sincerely hope the revised manuscript is now suitable for publication in International Journal of Molecular Sciences”. Finally, this manuscript has been edited by a a native English speaker to ensure accuracy and clarity.

Once again, thanks very much for taking your time to review this manuscript.

Reviewer 3 Report

This is a well-constructed and clearly described sequential approach to determine the role of melatonin in goose reproductive seasonality. The authors have substantiated through the introduction the basis for the study.  The scientific methods allow for the test of the hypothesis. The results support the conclusions.

l 444 change to slides

l 458 Delete And,

l 490 change is to are

Author Response

Response to Reviewer 3 Comments

Point 1: This is a well-constructed and clearly described sequential approach to determine the role of melatonin in goose reproductive seasonality. The authors have substantiated through the introduction the basis for the study. The scientific methods allow for the test of the hypothesis. The results support the conclusions.

Response 1: Thanks very much for taking your time to review this manuscript. We really appreciate the reviewer’s positive evaluation of our work. According to your advice, we amended the relevant part in manuscript. All of your questions were answered one by one, and hope that the correction will meet with approval.

Once again, thank you very much for your comments and suggestions.

Point 2: Line 444: change to slides

Response 2: We sincerely thank the reviewer for thoroughly examining our manuscript and providing very helpful comments to guide our revision. The manuscript has been revised according to the comment.

The sentence was modified as follows: “ Then, the slides were incubated with the primary antibody Anti-Mtnr (Mtnr, 1:1000, ab87639; Abcam, Cambridge, UK) at 4℃ overnight (Lines 471-473, page 15).”

Point 3: Line 458: Delete And,

Response 3: Thank you for your careful review. We are very sorry for the mistakes in this manuscript and inconvenience they caused in your reading. We have revised the text and hope that it is now clearer.

We modified the sentence as follows: “Slides of the pineal gland were dehydrated in ascending ethanol series then transferred to propylene oxide and embedded in Epon (SigmaAldrich) (Lines 486-488, page 15).”

Point 4: Line 490: change is to are

Response 4: Thank you for your suggestions. The insightful comments and suggestions are extremely helpful to improve and strengthen our manuscript. We have revised it correspondingly in the revised manuscript.

This sentence was modified as follows: “the data are the average of three independent experiments (Line 518, page 16).”

Thank you once again for your time in reviewing the manuscript. The resubmitted manuscript has been supplemented with more context and discussion. References were also added accordingly. All of your questions were answered one by one, and hope that the correction will meet with approval.

Round 2

Reviewer 1 Report

No major issue

Reviewer 2 Report

The Authors replied to all the comments. Therefore, the paper can be accepted in its form.